# Validity and reproducibility of a food frequency questionnaire to determine dietary intakes among Lebanese athletes

**Nagham Sannan**◉*[⊛], **Tatiana Papazian**[⊛], **Zeina Issa, Nour El Helou**[⊛]

Laboratory of Human Nutrition, Nutrition Department, Faculty of Pharmacy, Saint Joseph University of Beirut, Beirut, Lebanon

⊛ These authors contributed equally to this work.
* nagham.sannan@net.usj.edu.lb

**Data Availability Statement:** All relevant data are located at ZENODO. DOI: 10.5281/zenodo. 13840852.

## Abstract

### Background and objective

Nutrition is a basic need for athletes; thus, adequate dietary intake is crucial for maintaining overall health, facilitating training adaptations and boosting athletic performance. Accurate dietary assessment tools are required to minimize the challenges faced by athletes. This study verifies the validity and reproducibility of a 157 item semi-quantitative food frequency questionnaire (FFQ) among Lebanese athletes. This is the only Arabic questionnaire in Lebanon that estimates food consumption for athletes which can also be used in Arabic speaking countries. There has been no previous validated food frequency questionnaire that estimated food consumption for athletes in Lebanon.

### Methods

A total of 194 athletes were included in the study to assess the validity of the food frequency questionnaire against four days dietary recalls by comparing the total nutrient intake values from the food frequency questionnaire with the mean values of four 24-hour dietary recalls using Spearman correlation coefficient and Bland Altman plots. In order to measure the reproducibility, the intra class correlation coefficients were calculated by repeating the same food frequency questionnaire after one month.

### Results

The intra-class correlation coefficient between the two-food frequency questionnaires ranged from average (0.739 for carbohydrates) to good (0.870 for energy (Kcal)), to excellent (0.919 for proteins) concerning macronutrients and ranged from average (0.688 for vitamin D), to excellent (0.952 for vitamin B12), indicating an acceptable reproducibility. Spearman's correlation coefficients of dietary intake estimate from the food frequency questionnaire and the four dietary recalls varied between 0.304 for sodium, 0.469 for magnesium to 0.953 for caloric intake (kcal). Bland-Altman plots illustrated a percentage of agreement ranging between 94.3% for fats to 96.4% for proteins.

**Funding:** The author(s) received no specific funding for this work.

**Competing interests:** The authors have declared that no competing interests exist.

## Conclusion

This food frequency questionnaire has a reliable validity and reproducibility to evaluate dietary assessments and is an appropriate tool for future interventions to ensure the adoption of adequate eating strategies by athletes.

## Introduction

Sports nutrition is a modern discipline that involves the implementation of nutritional fundamentals in order to improve athletic performance [1]. Optimal nutrition delays fatigue, prolongs training durations and allows faster recovery between training sessions [2]. Thus, nutrition is an essential component of any physical activity, and that is the reason why active individuals strive to follow adequate dietary guidelines for maximum health optimization and sports performance [3]. Athletes' adherence to training programs and eating strategies produce optimal performance during competitions and quick recovery after exercise sessions [4, 5]. Moreover, dietary interventions through proper nutrient and food intake estimations stimulate overall health and competitive performance among athletes [6, 7]. Appropriate nutritional advice and counseling are essential to understand this population's knowledge. In epidemiological studies, distinctive methods are used to determine individuals' nutritional intake [8, 9]; the food frequency questionnaire (FFQ) is a low expense tool that is widely adopted in these studies among a large population [10]. The analysis of FFQs as beneficial methods has contributed in estimating dietary intakes in equitable measurements of energy, macronutrients, micronutrients, and fluids in various subgroups of the population worldwide [11]. Since dietary habits vary greatly depending on the ethnic, social and cultural background of the specified participants, FFQs must be tailored and validated to target specific populations because the food items should reflect their dietary habits [12]. In 2016, Lebanon has witnessed the validation of two FFQs, one among pregnant women [13] and the other among children [14]. Two other FFQs were validated in 2019 among Lebanese adults respectively [15, 16]. Since FFQs can serve as a valid and accurate tool for assessing the nutritional intakes of athletes [17], the purpose of this research is to validate a previously developed FFQ among Lebanese athletes.

## Materials and methods

### 2.1 Study population

Our study sample comes from the sports teams of Lebanese Military organizations represented by sports teams of the Lebanese Army and Internal Security Forces where these organizations have teams of professional athletes representing all the geographical areas of Lebanon. These teams are dedicated to sports and they are rigorously trained for participation in national and international competitions and also represent the country in these events. Based on this, and after acquiring the necessary authorizations from these organizations, the process of calculating the sample size began by gathering the names of all athletes from all military sports teams (18 different teams). The total was 255 athletes, and they were all contacted by phone and invited to participate in our study. A sample of 200 out of the 255 athletes agreed to participate but the data of 6 athletes were excluded due to missing values. Hence, the final sample size for our study was 194 athletes. These athletes come from all Lebanese geographical districts that included the capital Beirut and regions in Mount Lebanon, North Lebanon, South Lebanon

and the Beqaa. The study protocol was approved by the Institutional Review Board of Saint-Joseph University of Beirut and its Ethics Committee (USJ-2018-68) and was conducted according to the declaration of Helsinki guidelines. After an informative session prior to the study, all participants gave their written consent to participate in our study. The study was conducted between May 14, 2019 and March 11, 2020 by trained dietitians and face to face interviews. The athletes included in our study had to meet inclusion criteria that require them to be Lebanese, between the ages of 18 and 50, previous participants in Lebanese Championships for at least the three previous years as competent athletes, and must not have any medical conditions or use any type of medications that might influence their food intake. One hundred ninety-four participants were included in the final pool of analysis.

## 2.2 Development of the FFQ

The quantitative 157-item FFQ was developed by a panel of researchers and nutrition instructors from the faculty staff at the nutrition department of the faculty of pharmacy at Saint-Joseph University of Beirut (USJ). This FFQ was previously tested, in a pilot study, during a face-to-face interview on 50 athletes from the sample [12, 13] to assess its comprehensibility and acceptability among the athletes. The number of food items in the FFQ included 157 items that contain Mediterranean foods and foods with Middle Eastern ingredients and were divided into 12 food sections, with ingredients highly representative of the Lebanese cuisine, as follows: cereals and bread, rice and grains, milk and dairy products, fruits and juices, vegetables, meats and alternatives, nuts, sweets, bakeries, salty snacks, oils and fats, and finally beverages and drinks, including alcohol [18]. The number and frequency of each portion consumed was recorded on a daily, weekly, monthly, or yearly basis [19]. Seasonal fruit items were also taken into account while determining the frequency of their consumption throughout the period of the year [19]. In order to calculate the total amount of daily food intake (Kcal/day), the frequency of consumption was converted to times per day and then multiplied by its reported portion size [20]. This study adapted and validated the FFQ for athletes knowing that it was previously validated among the Lebanese adult population [13, 15]. The total duration for filling the FFQ was estimated to be around 40 minutes [21]. Licensed sports dietitians completed face to face interviews where each participant was asked specific questions about his/her food routine. To simplify the participants' answers, we used standard local household units such as plates, bowls, and spoons of different sizes, plastic food models and food photos. The same FFQ was administered at baseline (FFQ1) and one month later (FFQ2).

## 2.3 Methodology

In order to ensure complete validation of the FFQ, we underwent the average procedure of collecting a four day 24h dietary recalls [DRs] as a suitable compromise between the scientific vigor and the practicability, for the purpose of determining energy, macronutrient and micronutrient intakes. Hence, interviews were planned on weekdays and weekends to detect intake distinction among the participants. These participants filled 24h DR of two days repeated twice within a one-month interval with the intention of obtaining a total of four non-consecutive 24h DRs [22]. In addition to this, the research team took the anthropometric measurements and filled basic socio-demographic and lifestyle questionnaires of all the participants. In a systematic review, Cade et al. (2002) claimed that the 24h DRs questionnaire are used as an accurate reference method against FFQs [12]. For this reason, the validity of the FFQ was determined by comparing energy and nutrient intakes estimated from the FFQ and the average of the four-day DRs. This method guarantees less measurement errors and more reliable and credible associations [23]. Flowcharts in "Fig 1" provide a representation of this

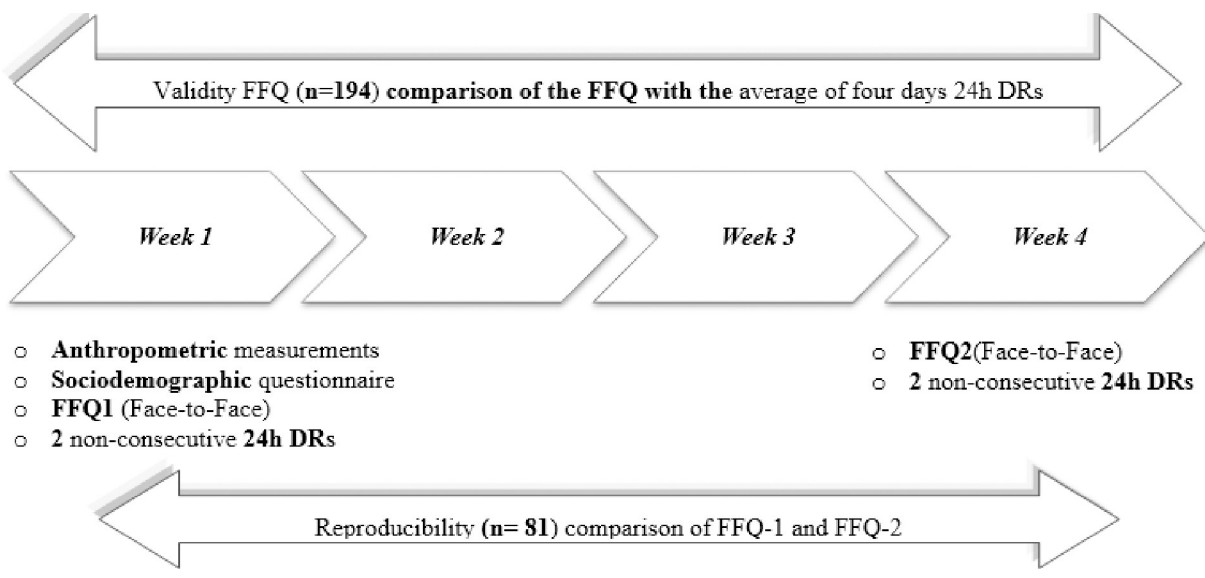

**Fig 1. Flowchart of the validation and reproducibility study among Lebanese athletes.**

information. Moreover, the five-step multiple-pass method declared by the United States Department of Agriculture was performed [24]. The primary reason for the use of this method is its accuracy and ease of food recall by the participants. The specific steps begin with an immediate list of foods consumed, potentially forgotten foods, followed by recording the meal times/ occasion for precision, then giving detailed quantities and ingredients, and a final review. Overall, participants needed 25–30 minutes to complete each 24h DR, accordingly with the recommendations [25].

The sample size chosen to test reproducibility was deliberately selected according to researchers in the field Cade et al. (2010) and Willet et al. (2012) who recommended a mini-mum percentage of 30% of the initial sample randomly chosen [8, 12]. In order to alleviate possible biases, questionnaires were filled by the same interviewer for all the athletes [22]. Interview-based questionnaires lead to faster completion and more consistent analysis com-pared to self-administered questionnaires [26]. To achieve the reproducibility of our tested tool, the licensed dietitians completed a second FFQ with the participants, one month later. The reproducibility of the FFQ was conducted among 81 participants who completed the ques-tionnaire twice by the dietitians within a 1-month time interval [25]. This interval was ade-quate for the participants to forget their previous responses, and was considered as a safe period to detect any significant and tangible changes in their dietary habits [12]. It should also be mentioned that exceeding a one-month interval might cause seasonal reporting biases [27].

## 2.4 Nutrient intake calculation

In order to process all the information provided by the FFQ, frequencies were converted into daily intake in grams. After that, analysis of the FFQ and the 24h DRs and the estimation of energy and nutrient intakes were performed respectively using Nutrilog® 2.3 software. The United States Department of Agriculture (USDA) database and the French food composition table CIQUAL were applied to provide nutrient values for all the international food items [28, 29]. Concerning Lebanese food recipes, the database of the American University of Beirut was used [30]. All three mentioned databases were incorporated within the Nutrilog® software.

**2.4.1 Relative validity of the food frequency questionnaire.** In order to accurately track energy and nutrient intakes for the FFQ and the four 24h DRs, means and standard deviations were calculated. The uniformity of the distribution was evaluated based on the usage of Kolmogorov-Smirnov test. The majority of nutrients underwent a skewed distribution, hence non-parametric tests were conducted. Variables of FFQs and 24h DRs were compared using the Wilcoxon signed-rank test to examine absolute differences between the 24h DRs and the FFQ1 and FFQ2 [31]. Furthermore, for the purpose of assessing an agreement between the two tools for total energy intake, macronutrients, and micronutrients, Bland Altman Plots were implemented. Spearman's correlation coefficient was used to test the reproducibility of the FFQ and the relationships between estimated macronutrients and micronutrients between the FFQs and 24h DRs [32]. The validity of this FFQ was determined by the correlation coefficients and Bland-Altman analysis by comparing the nutrient values of the initial FFQ (FFQ1) with the average nutrient of four 24-hour dietary recalls.

**2.4.2 Reproducibility of the food frequency questionnaire.** Nutrient intakes of both FFQs (FFQ1 and FFQ2) were assessed based on means and standard deviations. To measure the reproducibility of the FFQ, intra-class correlation coefficients (ICC) were calculated between values attained in the two FFQs. It was then evaluated using the following classification: weak reliability ($<0.5$), moderate reliability (0.5 to 0.75), good reliability (0.75 to 0.90), and excellent reliability ($>0.9$). The SPSS statistical software was applied to complete all statistical analyses (SPSS for windows, version 24.0, USA). A p-value less than 0.05 was interpreted as significant [33].

## Results

### 3.1 Socio-demographic and anthropometric characteristics of the population

The final sample included one hundred and ninety-four athletes, as shown in Table 1; the majority of the participants are males and their age range were between 21 and 47 years old with a mean of 30.05 years. Regarding the educational level, more than half of the population achieved secondary education or above and this is considered an acceptable academic level to have basic information and common knowledge about their food intakes and portion sizes. Furthermore, the majority of the population consumes two to three meals a day (62.4%).

### 3.2 Reproducibility of the food-frequency questionnaire

The reproducibility study involved eighty-one athletes. The results presented in Table 2, showed an ICC between the two FFQ measurements that ranged from average (0.739) for carbohydrates to good (0.870) for energy (Kcal), to excellent (0.919) for proteins concerning macronutrients, and ranged from average (0.688) for vitamin D to good (0.804) for vitamin B6 to excellent (0.952) for vitamin B12, (0.940) for sugar, and 0.917 for polyunsaturated fatty acids. These results indicate acceptable reproducibility of the tested tool.

### 3.3 Relative validity of the food-frequency questionnaire

The validation of the FFQ was assessed by comparing the intakes of energy, macronutrients, and micronutrients to four 24h DRs [34, 35]. Table 3 presents descriptive statistics and spearman correlations between nutrient intakes derived from the FFQ and the 24h DRs showing that FFQ's intakes are higher than 24h DRs ranging from 0.574 for proteins, 0.625 for fat, 0.953 for energy, 0.643 for vitamin E, and 0.557 for vitamin B12. According to the normality test, all variables in this case are skewed; hence, Wilcoxon signed rank test was executed and

Table 1. Characteristics of the study sample (n = 194).

| Characteristic | | Frequency | Percentage |
|---|---|---|---|
| Gender | Male | 183 | 94.30% |
| | Female | 11 | 5.70% |
| Civil status | Single | 85 | 43.80% |
| | Married | 22 | 11.30% |
| | Engaged | 84 | 43.30% |
| | Divorced | 2 | 1.00% |
| | Widow | 1 | 0.50% |
| Self- reported perceived financial status* | Good | 20 | 10.30% |
| | Medium | 94 | 48.50% |
| | Careful with expenses | 44 | 22.70% |
| | Critical | 27 | 13.90% |
| | No answer | 9 | 4.60% |
| Level of education | No education | 1 | 0.50% |
| | Primary | 25 | 12.90% |
| | Lower Secondary | 71 | 36.60% |
| | Secondary | 29 | 14.90% |
| | Baccalaureate | 43 | 22.20% |
| | Bachelor Degree | 22 | 11.30% |
| | Master Degree | 1 | 0.50% |
| | No answer | 2 | 1.00% |
| Number of meals consumed per day | 1–2 | 34 | 17.50% |
| | 2–3 | 121 | 62.40% |
| | 4–5 | 39 | 20.10% |

*Prior to the 2019 Lebanese financial crisis

demonstrated a significant difference between the means of DRs and FFQ (p<0.05). According to Spearman's test, there is a significant correlation between the FFQ and 24h DRs with a coefficient ranging between 0.304 for sodium, 0.469 for magnesium, 0.686 for vitamin A, 0.722 for carbohydrates (g/kg) and 0.953 for energy which is considered as a positive correlation ranging from weak to high. Cross-classification analysis revealed that about 92.8% of the participants were in the same quartile for vitamin A and 94.3% for polyunsaturated fatty acids.

## 3.4 Bland-Altman analysis

The graphical representation Bland-Altman plot illustrates the study of the agreement of the nutrients derived from the FFQ and the 24h DRs. The average difference between the two methods (FFQ and 24h DRs) is represented by the solid line, while the distance between the mean of the difference ± two standard deviations is designated by the hatched lines. The variation between positive and negative values demonstrated that nutritional intakes were both over and under estimated by the FFQ compared to 24h DRs which is considered to be the limit of agreement. The Bland-Altman analysis of energy, carbohydrates, fats, proteins, iron, and fatty acid intake is represented in "Fig 2" with a homogeneous dispersion above and below zero in most plots. The Bland-Altman method has proved to be the most effective in determining absolute validity by estimating mean agreement and the limits of agreement. Mean agreement indicates the average of the difference between the assessment tools that aid in determining food intake [36, 37]. This particular value was almost the same with the two

Table 2. Reproducibility study: Mean daily energy and nutrient intakes and the intra-class coefficient for the comparison between FFQ1 and FFQ2 in Lebanese athletes.

| Parameters | FFQ1 Mean ± SD | FFQ2 Mean ± SD | ICC (with 95% CI) |
|---|---|---|---|
| Energy (kcal) | 2526.72±587.68 | 2707.57±527.77 | 0.870 [0.798–0.916] |
| Carbs (%) | 55.81±10.00 | 57.80±10.26 | 0.857 [0.778–0.908] |
| Carbs(g) | 4.46±1.55 | 6.21±2.21 | 0.739 [0.594–0.832] |
| Sugars (g) | 27.29±8.06 | 30.82±9.12 | 0.940 [0.906–0.961] |
| Fat (%) | 32.19±7.53 | 35.60±6.70 | 0.825 [0.728–0.887] |
| Polyunsaturated Fatty Acids (g) | 18.63±6.11 | 18.01±8.18 | 0.917 [0.871–0.947] |
| Proteins (%) | 20.32±5.10 | 22.89±5.31 | 0.889 [0.827–0.929] |
| Proteins (g) | 1.61±0.59 | 1.74±0.61 | 0.919 [0.874–0.948] |
| Vitamin A(μg) | 311.16±224.68 | 317.85±201.29 | 0.885 [0.821–0.926] |
| Vitamin D (mg) | 11.49±4.19 | 10.79±2.73 | 0.688 [0.515–0.799] |
| Vitamin E (mg) | 11.11±5.33 | 11.78±5.67 | 0.942 [0.909–0.962] |
| Vitamin C (mg) | 88.11±46.23 | 103.24±47.93 | 0.930 [0.892–0.955] |
| Vitamin B6 (mg) | 2.55±0.81 | 2.69±0.93 | 0.804 [0.696–0.874] |
| Folate (μg) | 304.54±174.21 | 301.27±149.51 | 0.873 [0.802–0.918] |
| Vitamin B12 (mg) | 7.55±7.63 | 6.49±7.90 | 0.952 [0.925–0.969] |
| Magnesium (mg) | 377.56±153.68 | 384.12±159.31 | 0.910 [0.860–0.942] |
| Calcium (mg) | 935.10±296.46 | 976.84±258.59 | 0.905 [0.853–0.939] |
| Potassium (mg) | 3669.40±1327.78 | 3579.78±1359.59 | 0.897 [0.840–0.934] |
| Sodium (mg) | 6022.15±1345.10 | 6189.49±1445.99 | 0.949 [0.921–0.967] |
| Iron (mg) | 25.02±7.81 | 22.96±9.55 | 0.844 [0.757–0.899] |
| Zinc (mg) | 11.59±5.11 | 11.39±5.02 | 0.877 [0.809–0.921] |
| Selenium (mg) | 132.74±46.79 | 135.87±50.92 | 0.906 [0.854–0.939] |

assessment tools (the FFQ and the 24h DR), which shows a good degree of absolute validity. As for the nutrient intakes, the Bland- Altman analysis illustrated a percentage of agreement ranging between 94.3% for fats to 96.4% for proteins.

## Discussion

The objectives of this study were to examine the validity and the reproducibility of the FFQ administered among a group of 194 Lebanese athletes to assess their dietary intake. The tested tool has proved to be the most practical and cost-effective method in order to assess diets in large-scale nutritional epidemiological studies [38]. To our knowledge, this is the first study validating a culture-driven tool to assess the food consumption of athletes from Lebanon and Arab countries. This reliable tool included 157 food items that englobe the culinary back-ground of the Lebanese population and reflects their nutrition. The aim of the FFQ is to estab-lish a direct relationship between food intake and the athletes' condition which is highly affected by the quality and quantity of their meals [39]. The information derived from this FFQ allows us to know more about the dietary patterns and nutritional intakes of athletes which differs from the dietary pattern of the general population, knowing that athletes have different dietary consumptions that align with their respective training and physical activity.

Our current study validated the FFQ against 24-hour dietary recalls as required by literature in all epidemiological studies [40]. Nevertheless, some authors would prefer the collection of at least two 24h DRs due to the fact that this type of recall relies immensely on memory and is influenced by day-to-day and seasonal variation which is distributed along a definite time frame. Hence, in our study, to be more accurate and alleviate bias, four 24h DRs were collected

**Table 3. Validation study: FFQ and four 24h DRs.** Comparison of mean daily intake of energy and nutrients.

| Parameters | 24h DR Mean ± SD | FFQ Mean ± SD | p-value [a] | Relative Difference (%) [b] | Correlation Coefficient [c] |
|---|---|---|---|---|---|
| Energy (kcal) | 2359.20±597.48 | 2548.55±667.42 | <0.001 | 8.03% | 0.953 |
| Carbohydrates (%) | 49.81±8.58 | 54.75±9.72 | <0.001 | 9.92% | 0.868 |
| Carbohydrates (g/kg) | 3.78±1.29 | 4.13±1.62 | <0.001 | 9.35% | 0.722 |
| Sugar (g) | 25.40±9.62 | 28.03±7.81 | <0.001 | 10.35% | 0.511 |
| Fats (%) | 30.38±6.25 | 32.76±7.58 | <0.001 | 7.84% | 0.625 |
| Polyunsaturated fatty acids(g) | 12.55±5.83 | 19.23±6.04 | <0.001 | 53.22% | 0.573 |
| Proteins (%) | 19.74±5.14 | 20.85±6.19 | 0.002 | 5.63% | 0.574 |
| Proteins (g/kg) | 1.48±0.55 | 1.58±0.58 | <0.001 | 6.56% | 0.636 |
| Vitamin A (µg) | 253.29±284.57 | 335.71±228.93 | <0.001 | 32.54% | 0.686 |
| Vitamin D (µg) | 9.61±2.59 | 11.21±3.90 | <0.001 | 16.55% | 0.331 |
| Vitamin E (mg) | 7.66±6.20 | 11.28±6.22 | <0.001 | 47.21% | 0.643 |
| Vitamin C (mg) | 87.58±58.78 | 102.25±60.65 | <0.001 | 16.74% | 0.776 |
| Vitamin B6 (mg) | 2.02±1.15 | 2.62±1.05 | <0.001 | 29.25% | 0.809 |
| Folate (µg) | 290.18±149.64 | 314.20±180.01 | 0.001 | 8.28% | 0.720 |
| Vitamin B12 (mg) | 4.94±12.45 | 7.93±12.35 | <0.001 | 60.56% | 0.557 |
| Magnesium (mg) | 309.00±132.67 | 377.63±163.13 | <0.001 | 22.21% | 0.469 |
| Calcium (mg) | 667.49±300.90 | 946.07±309.24 | <0.001 | 41.74% | 0.637 |
| Potassium (mg) | 2905.06±1065.72 | 3580.19±1326.72 | <0.001 | 23.24% | 0.426 |
| Sodium (mg) | 4701.94±1161.34 | 6002.03±1409.63 | <0.001 | 27.65% | 0.304 |
| Iron (mg) | 16.57±6.47 | 25.06±9.37 | <0.001 | 51.26% | 0.353 |
| Zinc (mg) | 8.75±4.42 | 10.84±5.38 | <0.001 | 23.82% | 0.542 |
| Selenium (µg) | 102.76±72.88 | 140.10±65.44 | <0.001 | 36.33% | 0.776 |

[a] Wilcoxon signed rank test were used to compare values obtained from both FFQ and 4-24h DRs

[b] Relative difference = [(FFQ– 24h DR)/24h DR]x100.

[c] Spearman correlation coefficient were used to assess the correlation between variables with a p-value < 0.05 considered as significant

for each athlete by the same interviewer [41]. It must be noted that the one-month interval between repeated measurements is sufficient to reduce dietary changes over time as well as the memory of previous answers [12, 42]. As a matter of fact, a time interval more than one month and up to three months could lead to the emergence of a seasonality bias where some foods consumed by athletes might change between seasons due to the availability of certain types of fruits and vegetables in each season [43]. Despite this, previous studies had time intervals of two weeks [44], three weeks [45, 46], four weeks [47, 48], four to six weeks [49] and six weeks [50, 51]. In our study, validity was assessed using correlation coefficients (CCs) by comparing energy and nutrient intakes estimated from the FFQ and the average of the four-day DRs [52, 53]. According to Willet (1998) and Cade et al. (2002), the best obtainable values for CCs in dietary validation studies are between 0.5 and 0.7. In our study, the correlation coefficients were above 0.5 for the majority of nutrients: 0.953 for energy (kcal), 0.868 for carbohydrates, 0.625 for fats (%), 0.574 for proteins (%), 0.776 for vitamin C and 0.686 for vitamin A. Our data indicate a relatively high correlation level between the FFQ and the reference method, and this can be attributed to athletes' increased focus and further attention to details concerning their foods and eating habits in comparison with the usual population. In general, athletes report their diet accurately since they are aware of the implications of their food intakes on their performance. In comparison with many other studies, a few have been performed on

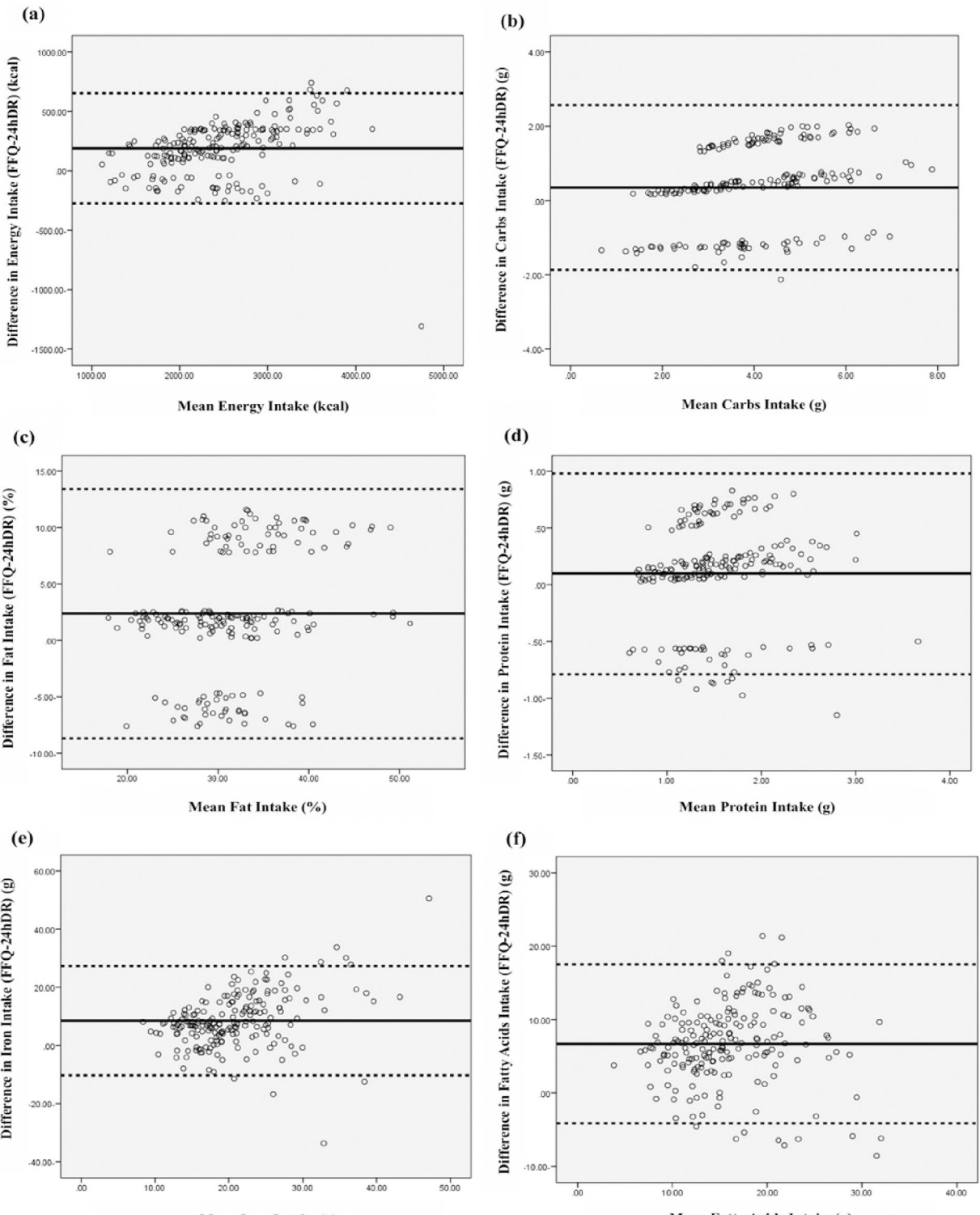

**Fig 2. Bland-Altman plots representing the agreement between the average of nutrients intakes measured from the FFQ1 and the average of four 24hDRs.** (a) Bland-Altman analysis for energy intake; (b) Bland-Altman analysis for carbs intake; (c) Bland-Altman analysis for fat intake; (d) Bland-Altman analysis for protein intake; (e) Bland-Altman analysis for iron intake; (f) Bland-Altman analysis for fatty acids intake.

athletes; thus, this study shows CC values within a successful range (0.636 of proteins (g/kg) →0.953 energy) regarding macronutrients and (0.3→0.809) regarding micronutrients. According to Morel et al. in a French-Canadian study conducted in 2018, the values ranged between 0.56 and 0.88 [54]. Moreover, another study published by Fallaize et al. (2014) showed that the CCs ranged between 0.65 and 0.9; which are considered to be substantially higher than those found in an earlier study [55]. To further ensure the highly acceptable CCs in our own study, two other examples were taken into consideration; the first one involved Cheng et al. (2020) that recorded a CC of 0.63 for carbohydrates, 0.21 for fats, and 0.34 for proteins, 0.62 for vitamin C and 0.55 for vitamin A [56]. The second study by Sasaki et al. (2018) noted a median CC of 0.3 for males and 0.32 for females [57]. As a result, our range has proved to have a higher CC than in those two mentioned studies. In our study, there are some factors that have contributed to the high validity and correlation compared to others. First, the comprehensive food list positively affected the results. Second, the method of administration of the questionnaire in which data is collected by a professional dietitian with the same exact athlete mitigated the variability. Third, the personal interview presented spontaneous and accurate responses from the participants. Finally, the type of questions facilitated effective communication and contributed to the validity of the results. Moreover, stronger correlations were withheld upon the exclusions of participants who tend to either under or overestimate their total energy intakes, and who suffer from a certain health condition that could have an impact on their nutritional intake or health status. As for the arrangement of the intake levels, the quartile notation was most suitable; this study is appropriate and compatible with results previously reported by Zaragoza et al. (2017) in which more than 84% of the subjects were grouped in the same quartile [58]. We noted that there was a consistent overestimation of the FFQ to most of the nutrient intakes reaching 5.6% for protein, 7.81% for fats, 47.2% for vitamin E and 60.6% for vitamin B12 with a difference of 8% for energy. This variation between the FFQ and 24h DRs could be mainly due to the fact that athletes tend to eat bigger portions and each one of them focuses on his food intake depending on the season and competitions. Moreover, this overestimation can be related to changes in athletes' appetites in different days due to variations in their training programs, such as during competitions or during training phase. Another possible reason for this overestimation is the number of food items that makes it difficult for participants to recall their exact yearly consumption while filling the FFQ with respect to the 24h DRs [48, 59] as Pinto et al.(2010) proposed that this overestimation may be due to difficulties in accurately estimating the portion size consumed [60, 61]. Consequently, the Bland Altman analyses were performed in order to demonstrate the compliance between the FFQ and the 24h DRs by showing the difference between the two methods against their averages. A mean of differences that is close to zero implies a more comprehensive agreement between the two tools. Hence, this FFQ displayed a very good validity compared to the 24h DRs [62]. Consequently, the range of statistical tests demonstrated that this cultural FFQ is a very useful tool for dietary assessments within the athletic community when compared to four 24 h dietary recalls. We can also use it to identify potential problems, prevent deficiency and provide athletes with effective nutritional strategies to achieve their sporting goals in a healthy manner.

The reproducibility of the FFQ was determined by administering the same FFQ twice, with a one-month interval between administrations. This reproducibility was found to be good since the ICC from our study ranged from 0.825 to 0.889 for macronutrients and acceptable from 0.688 to 0.952 for micronutrients. We observed that energy intake and macronutrient intakes in FFQ2 were slightly higher than those in FFQ1, for example energy intake was (2707.57±527.77 kcal) in FFQ2 versus (2526.72±587.68 kcal) in FFQ1, carbohydrates intake was (6.21±2.21 g) in FFQ2 versus (4.46±1.55 g) in FFQ1, and protein intake was (1.74±0.61) in

FFQ2 versus (1.61±0.59) in FFQ1. This can be explained by the increase in the level of engagement of the participants regarding their answers to the questions as well as their desire to give accurate answers in completing FFQ2. This can also be interpreted by the fact that participants became more familiar with the same dietitian who administered the same questionnaire twice. Compared to other studies, Ishikawa-Takata et al.'s (2021) results for Japanese athletes showed an ICC of 0.588 for carbohydrate, 0.619 for energy and 0.752 for vitamins [63]. Moreover, a study of the validity and reproducibility of a culture–specific FFQ developed in Lebanon demonstrated an ICC of 0.44 for protein, 0.67 for fat and 0.85 for energy [32]. As a result, these observations indicate that our FFQ has a high reproducibility; this might be explained by the fact that these athletes included in our study are subject to regular nutritional follow ups in a structured environment controlled by professional sport dietitians.

Based on our findings, we can conclude that this FFQ has good validity and reproducibility and can be considered an efficient and reliable dietary assessment tool to evaluate the dietary intakes of high performing athletes. Hence, this FFQ can be used in future researches and studies to assess athletes' food group intakes and to identify the nutritional habits and behaviors of these athletes. Therefore, sports experts and dietitians can provide athletes with effective nutritional strategies that help promote training adaptations and allows them to achieve their sports goals and improve their performance.

## Strengths and limitations

We acknowledge some inevitable limitations concerning this research. The extended list of 157 food items might have biased the answers of the participants and have led to overestimations [64]. As for the validation of the FFQ, the 24h DR was chosen to be the reference method which could be seen a possible weakness since both the FFQ and the 24h DR have related measurement errors including memory bias, misreporting of portion sizes and cooking techniques, and other seasonal variations [65, 66]. In addition, the heterogeneity of the sample of athletes, derived from 18 professional sports teams, can influence the results. A serious potential error that is common to self-reported dietary methods is the estimation of nutrients based on food composition tables. The nutritional composition of food varies depending on the season, location of production, growing conditions, storage, processing and cooking techniques. A lot of these mentioned factors are not taken into consideration in the food composition tables. It must be noted that determining the nutritional composition of food intake is a significant challenge, which is the main reason to continuously work towards improving the accuracy of food composition databases.

On the other hand, it is worth mentioning that the current study's major strength was the comprehensive approach utilized to establish the FFQ items that are suitable and in line with the food culture of the Lebanese athletes. The fact that the FFQ was interviewer-administered and not self-administered improved accuracy by allowing for immediate clarification of any misunderstandings and reduced inaccuracies. It also allowed more control by ensuring standardized portion sizes and intake frequencies and reducing the risk of misinterpretations by participants. This FFQ includes food items that are related to the Mediterranean culture in the Arab world, making it a reliable tool in Arabic language that can be used by Arab-speaking athletes in the Arab world. Moreover, the sample size exceeds the average number of participants in other validation studies since our study had a high participation rate. Another strength for this study is the statistical methodology applied. A recent review conducted by Lombard et al. (2015) claimed that the percentage of statistical tests usually used in validation studies range between one and three. Nevertheless, we applied all appropriate methods while developing a methodological framework that is considered adequate to transmit logical

insights into various factors of validity. Thus, when various statistical analyses are administered, a better reproducibility and validity of our tools are found vividly.

When it comes to critical research issues, time constraints, financial factors, and participant burdens are clearly found which is why future investigations in this particular field should develop online questionnaires for a better assessment of the nutritional intakes. Also, using computerized systems in epidemiology has the ability to favor the study by permitting faster responses, mitigating the bias of incomplete data using automated in controls which eases the estimation of food portions through illustrative models.

## Conclusion

The purpose of this study was to adopt a 157-item semi-quantitative FFQ that was already developed and validated among Lebanese adults and Lebanese pregnant women, to evaluate its validity and reproducibility among Lebanese Athletes. In conclusion, we attempt to offer a valid and reliable FFQ that would be beneficial in estimating nutrient intakes for the purpose of developing appropriate strategies to progress the diets of the athletes needing a thorough investigation of their dietary habits. This particular FFQ is also appropriate for trials that assess the impact of nutritional interventions on dietary practices. In order to limit the FFQ's administration time and to mitigate the length of the listed items given that they tend to cause confusion and overestimation of consumption among athletes, future work and focus must be done to develop a smaller yet efficient computerized version of this FFQ, taking into consideration the special methodological considerations for athletes including serving sizes, having snacks, water and beverage consumption, supplement use and weight control.

## Supporting information

**S1 Data. Data validity.**
(XLSX)

**S2 Data. Data reliability.**
(XLSX)

**S1 File. Food frequency questionnaire.**
(DOCX)

**S2 File. Food frequency questionnaire-standard English version.**
(DOCX)

## Acknowledgments

The authors would like to thank all the participants whose contribution helped in the success of this research. We would also like to thank Mrs. Hanine Abou Antoun for her contribution and help in statistics and Mrs. Steffi Younes for her contribution to the research.

## Author Contributions

**Conceptualization:** Nagham Sannan, Nour El Helou.

**Data curation:** Nagham Sannan, Zeina Issa, Nour El Helou.

**Formal analysis:** Nagham Sannan, Nour El Helou.

**Investigation:** Nagham Sannan, Tatiana Papazian, Zeina Issa, Nour El Helou.

**Methodology:** Nagham Sannan, Tatiana Papazian, Zeina Issa, Nour El Helou.

**Project administration:** Nagham Sannan, Nour El Helou.

**Supervision:** Nour El Helou.

**Validation:** Nagham Sannan, Tatiana Papazian, Zeina Issa, Nour El Helou.

**Writing – original draft:** Nagham Sannan.

**Writing – review & editing:** Nagham Sannan, Tatiana Papazian, Nour El Helou.

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
