## [Decision Letter · Decision Letter 0]

9 Jul 2024

PONE-D-24-19959Validity and reproducibility of a food frequency questionnaire to determine dietary intakes among Lebanese athletesPLOS ONE

Dear Dr. Sannan,

Thank you for submitting your manuscript to PLOS ONE. After careful consideration, we feel that it has merit but does not fully meet PLOS ONE’s publication criteria as it currently stands. Therefore, we invite you to submit a revised version of the manuscript that addresses the points raised during the review process.

Dear Dr. Sannan,

About the paper: Validity and reproducibility of a food frequency questionnaire to determine dietary intakes among Lebanese athletes

Some major revisions to the manuscript have been suggested within peer review.

The title and methodology of the study indicate that the research was conducted on the behavior of Lebanese athletes. However, if the intention is to utilize the questionnaire for the general Lebanese population, there is a high probability of selection bias, as the sample cannot be generalized to the entire Lebanese society. It is evident that athletes represent a relatively small sample of each society. Furthermore, it is clear that they engage in a significantly higher level of physical activity than the general population, which inevitably leads to a different dietary pattern. Consequently, the results of this study cannot be generalized to the wider Lebanese society.

Please clarify whether the results of this study will be important only in the athletes community?

We look forward to receiving your revised manuscript.

Kind regards,

Zahra Cheraghi, Ph.D

Academic Editor

PLOS ONE

Journal Requirements:

Additional Editor Comments:

Dear Dr. Sannan,

About the paper: Validity and reproducibility of a food frequency questionnaire to determine dietary intakes among Lebanese athletes

Some major revisions to the manuscript have been suggested within peer review.

The title and methodology of the study indicate that the research was conducted on the behavior of Lebanese athletes. However, if the intention is to utilize the questionnaire for the general Lebanese population, there is a high probability of selection bias, as the sample cannot be generalized to the entire Lebanese society. It is evident that athletes represent a relatively small sample of each society. Furthermore, it is clear that they engage in a significantly higher level of physical activity than the general population, which inevitably leads to a different dietary pattern. Consequently, the results of this study cannot be generalized to the wider Lebanese society.

Please clarify whether the results of this study will be important only in the athletes community?

Reviewers' comments:

Reviewer's Responses to Questions

**Comments to the Author**

1. Is the manuscript technically sound, and do the data support the conclusions?

Reviewer #1: Yes

Reviewer #2: Yes

2. Has the statistical analysis been performed appropriately and rigorously? 

Reviewer #1: Yes

Reviewer #2: No

3. Have the authors made all data underlying the findings in their manuscript fully available?

Reviewer #1: No

Reviewer #2: No

4. Is the manuscript presented in an intelligible fashion and written in standard English?

Reviewer #1: Yes

Reviewer #2: Yes

5. Review Comments to the Author

Reviewer #1: 1. Authors should refer to the methods used to measure validity and reliability in the abstract method.

2. The authors stated in the abstract results that the amount of ICC is excellent. According to various sources, ICC value above 0.9 is excellent, 0.75 to 0.9 is good, 0.5 to 0.75 is average, and below 0.5 is poor. Therefore, the authors should not introduce the value of 0.688 or 0.739 or 0.870 as excellent.

3. The logic of calculating the sample size of 194 athletes should be explained in the method section.

4. In lines 196-202, the authors should not use the term excellent reproducibility because not all ICCs are above 0.9.

5. As an educational note, in scientific writing all tables and figures should be moved to the end of the last reference.

6. In lines 206 to 214, the author reports Spearman's correlation coefficient. The value of this index is between -1 and +1. Please state the reporting range of Spearman's correlation coefficient on this scale.

7. I do not understand the concept of seasonality bias in line 258 of the discussion. Please explain it.

8. In the discussion of the article, the author should not simply compare his findings with others. In the discussion, they should mention the reasons for the differences between their findings and others. Please consider this point in rewriting the discussion.

Reviewer #2: attached

6. PLOS authors have the option to publish the peer review history of their article (what does this mean?). If published, this will include your full peer review and any attached files.

Reviewer #1: **Yes: **Fatemeh Shahbazi

Reviewer #2: No

---

## [Author Response · Author response to Decision Letter 0]

7 Aug 2024

Editor’s question and comments:

The title and methodology of the study indicate that the research was conducted on the behavior of Lebanese athletes. However, if the intention is to utilize the questionnaire for the general Lebanese population, there is a high probability of selection bias, as the sample cannot be generalized to the entire Lebanese society. It is evident that athletes represent a relatively small sample of each society. Furthermore, it is clear that they engage in a significantly higher level of physical activity than the general population, which inevitably leads to a different dietary pattern. Consequently, the results of this study cannot be generalized to the wider Lebanese society.

Please clarify whether the results of this study will be important only in the athlete’s community?

Response: 

The purpose of our research is to validate the food frequency questionnaire (FFQ) among the specific community of Lebanese athletes. This validation will enable us to use it as a tool for assessing athletes’ food consumption in Lebanon and other Arab countries, as well as for studying the relationship between nutrition and performance. Validated questionnaires are essential in research studies, and this validated FFQ is the only tool that can assess the nutritional status of athletes in Lebanon and other Arabic-speaking countries. 

It is important to note that the same food frequency questionnaire (FFQ) has been previously validated and published for use among Lebanese adults and Lebanese pregnant women (13, 15), so it can be utilized within the general Lebanese adult population, and pregnant women. For this present study, we chose to administer and validate the same food frequency questionnaire for Lebanese athletes because it includes 157 food items that cover the culinary background of the Lebanese population, thus reflecting their typical nutrition. The information derived from this food frequency questionnaire allows us to obtain insights into the dietary patterns and nutritional intakes of Lebanese and Arab-speaking athletes which differs from those of the general population. Athletes have unique dietary requirements that align with their training and physical needs.

Reviewer 1

We thank the reviewer for the insightful comments and valuable feedback, and we appreciate your time and efforts in reviewing our work. Please find below our replies to your comments.

1. Authors should refer to the methods used to measure validity and reliability in the abstract method.

We edited our abstract in response to your comment and included the methods described in the “methods section” of our abstract as follows:

To assess the validity of the food frequency questionnaire (FFQ), we compared total nutrient intake values from the FFQ with the average nutrient intake values of four 24 h dietary recalls using the Spearman correlation coefficient and Bland-Altman plots.

To measure the reliability of the FFQ, we calculated intra-class correlation coefficients by administering the same FFQ twice, with a one-month interval between administrations.

2. The authors stated in the abstract results that the amount of ICC is excellent. According to various sources, ICC value above 0.9 is excellent, 0.75 to 0.9 is good, 0.5 to 0.75 is average, and below 0.5 is poor. Therefore, the authors should not introduce the value of 0.688 or 0.739 or 0.870 as excellent.

We replaced the term “excellent” as follows:

“The intra-class correlation coefficient between the two-food frequency questionnaires ranged from average (0.739) for carbohydrates to good (0.870) for energy (Kcal), to excellent (0.919) for proteins concerning macronutrients and ranged from average (0.688) for vitamin D, to excellent (0.952) for vitamin B12, indicating an acceptable reproducibility.”

3. The logic of calculating the sample size of 194 athletes should be explained in the method section.

We added the explanation of the sample size in the method section.

The process of calculating the sample size began by gathering the names of all athletes from all military sports teams. The total was 255 athletes, and they were all invited to participate in our study. A sample of 200 out of the 255 athletes agreed to participate but the data of 6 athletes were excluded due to missing values. Hence, the final sample size for our study was 194 athletes.

4. In lines 196-202, the authors should not use the term excellent reproducibility because not all ICCs are above 0.9.

We replaced the term “excellent” as follows in lines 212 – 220 in the “Revised Manuscript with Track Changes” file:

“The results presented in table 2 showed an ICC between the two FFQ measurements that ranged from average (0.739) for carbohydrates to good (0.870) for energy (Kcal), to excellent (0.919) for protein concerning macronutrients, and ranged from average (0.688) for vitamin D to good (0.804) for vitamin B6 to excellent for vitamin B12 , (0.940) for sugar, and 0.917 for polyunsaturated fatty acids. These results indicate an acceptable reproducibility of the tested tool.”

5. As an educational note, in scientific writing all tables and figures should be moved to the end of the last reference.

The tables and figures are moved to the end. Done. 

6. In lines 206 to 214, the author reports Spearman's correlation coefficient. The value of this index is between -1 and +1. Please state the reporting range of Spearman's correlation coefficient on this scale.

The coefficients have been reported between -1 and +1 as per your suggestion in lines 222 – 229 of the “Revised Manuscript with Track Changes” file:

“The validation of the FFQ was assessed by comparing the intakes of energy, macronutrients, and micronutrients to four 24h DRs [34, 35]. Table 3 presents descriptive statistics and Spearman correlations between nutrient intakes derived from the FFQ and the 24h DRs showing that FFQ’s intakes are higher than 24h DRs ranging from 0.574 for proteins, 0.625 for fat, 0.953 for energy, 0.643 for vitamin E, and 0.557 for vitamin B12. According to the normality test, all variables in this case are skewed; hence, Wilcoxon signed rank test was executed and demonstrated a significant difference between the means of DRs and FFQ (p<0.05).”

7. I do not understand the concept of seasonality bias in line 258 of the discussion. Please explain it.

The concept of seasonality bias is prevalent in Lebanon due to the country’s weather that is characterized by the existence of four changing seasons which produce different kinds of fruits and vegetables in each of these seasons. Hence, the type of foods consumed by the population changes between the seasons based on the availability of certain types of foods per season. An explanation has been added to the discussion in lines 279 – 280 of the “Revised Manuscript with Track Changes” file.

8. In the discussion of the article, the author should not simply compare his findings with others. In the discussion, they should mention the reasons for the differences between their findings and others. Please consider this point in rewriting the discussion.

We added the reasons and included them in the re-written discussion section. Done. 

Reviewer 2

We thank the reviewer for the insightful comments and valuable feedback, and we appreciate your time and efforts in reviewing our work. Please find below our replies to your comments.

To ensure the validity of a questionnaire, you must go through several stages and your work is incomplete

We passed through several stages to ensure the validity of the food frequency questionnaire (FFQ) and are mentioned in the Materials and methods section of the “Revised Manuscript with Track Changes” file as follows:

1. A panel of researchers and nutrition instructors who engaged in many discussions and meetings developed the FFQ and started by creating a list of the mostly consumed food items from the Lebanese cuisine and specifically by athletes. The number of food items in the FFQ included 157 items that contain Mediterranean foods and foods with Middle Eastern ingredients and were organized into 12 groups.

2. The FFQ was tested in a pilot study of 50 athletes to study the comprehensibility and acceptability of the questionnaire among them. Licensed sports nutrition dietitians conducted face-to-face interviews with these athletes to complete the questionnaire and then asked them about the FFQ’s comprehensibility and acceptability. To help athletes quantify the exact amount and obtain accurate information, portion sizes were illustrated with pictures or by using plastic food models and household measures (cups and spoons).

3. Face-to-face interviews were conducted on the sample of 194 athletes by licensed dietitians who took anthropometric measures, collected socio-demographic characteristics and personal information about athletes, and then completed the FFQ. The frequency of consumption and serving size was then recorded on a daily, weekly, monthly, yearly basis, or never. The reported frequency for each food item in the FFQ was converted to gram per day according to the Lebanese household foods measurements.

4. Four-24-hour dietary recalls were completed for each athlete. This is typically a three-stage process that starts by collecting a list of all foods and fluids consumed by athletes. The second step was conducted by gathering detailed descriptions of each food item such as their brands, their cooking methods and the portion sizes consumed. Finally, a review was conducted with the athletes to ensure that all foods and fluids have been accurately recorded.

5. The same FFQ was re-administered after one month by the same licensed sport dietitian who collected data from the same athletes to ensure less variability.

6. Data from the food frequency questionnaire and data from the mean intake of the four-24-hour dietary recalls were uploaded into Nutrilog®, a software designed to analyze dietary intakes and calculate the daily energy and nutrient intakes for athletes. The validity of this FFQ was then determined by the correlation coefficients test and Bland-Altman analysis by comparing the nutrients values of the initial FFQs (FFQ1) with the average nutrient of four-24-hour dietary recalls.

2- It’s related to the narrative of the questionnaire in question, not what the athlete consumes.

In order to assess the athletes’ nutrient intake and know about their nutritional habits and consumption, we had to validate a dietary assessment tool, the FFQ. We formulated this FFQ to include as much items as possible to allow us to gather information about the food consumption of athletes. The validation of this FFQ was conducted by comparing nutrient intakes with the mean values of four dietary recalls using the correlation coefficient and Spearman correlation. The dietary recalls (four 24-h recalls) allowed us to assess exactly what the athletes consumed, which was consistent with the items included in the FFQ. This validated FFQ tool of 157 items contains Middle Eastern and Mediterranean foods that represent the Lebanese and Middle-Eastern cuisine providing accurate data about the nutritional intake of athletes. 

Thank you once again for your valuable feedback

---

## [Editor Report · Decision Letter 1]

12 Aug 2024

PONE-D-24-19959R1Validity and reproducibility of a food frequency questionnaire to determine dietary intakes among Lebanese athletesPLOS ONE

Dear Dr. Sannan,

Thank you for submitting your manuscript to PLOS ONE. After careful consideration, we feel that it has merit but does not fully meet PLOS ONE’s publication criteria as it currently stands. Therefore, we invite you to submit a revised version of the manuscript that addresses the points raised during the review process.

**ACADEMIC EDITOR: **Dear Editor in Chief

The revised version of the paper (PONE-D-24-19959R1 : Validity and reproducibility of a food frequency questionnaire to determine dietary intakes among Lebanese athletes

) was submitted in accordance with the recommendations set forth by the reviewers. The original authors are responsible for reviewing the edited article once more to confirm or verify the changes made.

Best

We look forward to receiving your revised manuscript.

Kind regards,

Zahra Cheraghi, Ph.D

Academic Editor

PLOS ONE

---

## [Author Response · Author response to Decision Letter 1]

18 Aug 2024

We are thankful again for your valuable comments and feedback that improved our submission.

---

## [Decision Letter · Decision Letter 2]

23 Sep 2024

Validity and reproducibility of a food frequency questionnaire to determine dietary intakes among Lebanese athletes

PONE-D-24-19959R2

Dear Dr. Sannan,

We’re pleased to inform you that your manuscript has been judged scientifically suitable for publication and will be formally accepted for publication once it meets all outstanding technical requirements.

Kind regards,

Zahra Cheraghi, Ph.D

Academic Editor

PLOS ONE

Additional Editor Comments (optional):

Dear Editor

I hope this message finds you well. I am writing to formally accept the invitation to review the manuscript titled “ONE-D-24-19959R2: Validity and reproducibility of a food frequency questionnaire to determine dietary intakes among Lebanese athletes]. My final decision, based on the original reviewers' opinions, is acceptance.

Thank you once again for this opportunity. I look forward to contributing to the advancement of our discipline through this review process.

Best regards,

Reviewers' comments:

Reviewer's Responses to Questions

**Comments to the Author**

1. If the authors have adequately addressed your comments raised in a previous round of review and you feel that this manuscript is now acceptable for publication, you may indicate that here to bypass the “Comments to the Author” section, enter your conflict of interest statement in the “Confidential to Editor” section, and submit your "Accept" recommendation.

Reviewer #1: All comments have been addressed

2. Is the manuscript technically sound, and do the data support the conclusions?

Reviewer #1: Yes

3. Has the statistical analysis been performed appropriately and rigorously? 

Reviewer #1: Yes

4. Have the authors made all data underlying the findings in their manuscript fully available?

Reviewer #1: Yes

5. Is the manuscript presented in an intelligible fashion and written in standard English?

Reviewer #1: Yes

6. Review Comments to the Author

Reviewer #1: Dear authors

About article entitled "Validity and reproducibility of a food frequency questionnaire to determine dietary intakes among Lebanese athletes"Many thanks for your corrections.

In my opinion, your article can be published in Plos One

7. PLOS authors have the option to publish the peer review history of their article (what does this mean?). If published, this will include your full peer review and any attached files.

Reviewer #1: **Yes: **Fatemeh Shahbazi

---

## [Editor Report · Acceptance letter]

12 Oct 2024

PONE-D-24-19959R2 

PLOS ONE

Dear Dr. Sannan, 

I'm pleased to inform you that your manuscript has been deemed suitable for publication in PLOS ONE. Congratulations! Your manuscript is now being handed over to our production team.

Kind regards, 

on behalf of

Dr. Zahra Cheraghi 

Academic Editor

PLOS ONE